# Quality of Life for Head and Neck Cancer Patients: A 10-Year Bibliographic Analysis

**DOI:** 10.3390/cancers15184551

**Published:** 2023-09-14

**Authors:** Siti Nur Akmal Ghazali, Caryn Mei Hsien Chan, Marfu’ah Nik Eezamuddeen, Hanani Abdul Manan, Noorazrul Yahya

**Affiliations:** 1Diagnostic Imaging and Radiotherapy, CODTIS, Faculty of Health Sciences, National University of Malaysia, Jalan Raja Muda Aziz, Kuala Lumpur 50300, Malaysia; 2Clinical Psychology and Behavioural Health Programme, REACH, Faculty of Health Sciences, National University of Malaysia, Jalan Raja Muda Aziz, Kuala Lumpur 50300, Malaysia; 3Cancer Center MAKNA, Universiti Kebangsaan Malaysia Medical Centre, Cheras, Kuala Lumpur 56000, Malaysia; 4Functional Image Processing Laboratory, Department of Radiology, Universiti Kebangsaan Malaysia, Cheras, Kuala Lumpur 56000, Malaysia; 5Department of Radiology and Intervention, Hospital Pakar Kanak-Kanak (Children Specialist Hospital), Universiti Kebangsaan Malaysia, Jalan Yaacob Latif, Bandar Tun Razak, Kuala Lumpur 56000, Malaysia

**Keywords:** quality of life, head and neck cancer, bibliometric analysis, survivorship

## Abstract

**Simple Summary:**

This article aims to investigate the changing focus in quality of life (QoL) studies for head and neck cancer (HNC) patients and foster global collaborations. The specific objectives are to understand the unique challenges faced by HNC patients through an analysis of key terms used in recent studies. By examining existing literature and analyzing keywords, the authors seek to identify critical areas for future research and highlight important topics. The findings from this research have significant potential to benefit the research community by providing insights into the specific needs of HNC patients, guiding future studies, and facilitating collaboration among researchers and policymakers. Ultimately, this research aims to enhance our understanding and support for HNC patients, ultimately improving their QoL.

**Abstract:**

Head and neck cancers (HNCs) have a profound impact on patients, affecting not only their physical appearance but also fundamental aspects of their daily lives. This bibliometric study examines the landscape of scientific research pertaining to the quality of life (QoL) among head and neck cancer (HNC) patients. By employing data and bibliometric analysis derived from the Web of Science Core Collection (WOS-CC) and employing R-package and VOSviewer for visualization, the study assesses the current status of and prominent areas of focus within the literature over the past decade. The analysis reveals noteworthy countries, journals, and institutions that have exhibited notable productivity in this research domain between 2013 and 2022. Notably, the United States, the *Supportive Care in Cancer* journal, and the University of Pittsburgh emerged as the leading contributors. Moreover, there was a discernible shift, with an increasing focus on the significance of QoL within the survivorship context, exemplified by the emergence and subsequent peak of related keywords in 2020 and the subsequent year, respectively. The temporal analysis additionally reveals a transition towards specific QoL indices, such as dysphagia and oral mucositis. Therefore, the increasing relevance of survivorship further underscores the need for studies that address the associated concerns and challenges faced by patients.

## 1. Introduction

Around the world, head and neck cancer (HNC) is the sixth most prevalent cancer, with an estimated 660,000 new cases and 325,000 fatalities per year [1]. The total incidence of HNC continues to increase, with a projected 30% yearly increase by 2030 [2]. Treatment for individuals with HNC involves a comprehensive approach, and most of them will experience radiotherapy as the primary treatment or as an adjuvant to surgery. Still, radiation-related problems have a significant impact on quality of life (QoL) [3]. The analysis of patient issues following treatment revealed that patients’ requirements are numerous and diverse [4]. Importantly, unrecognized patient concerns may result in unmet patient needs, which will likely have an impact on patient survival [5]. The impact of nutritional changes on patients’ lives, including weight loss, dysphagia, xerostomia, and taste changes; the debilitating effects of persistent fatigue; the unpreparedness for and distress from the radiotherapy mask; and attempts to maintain a normal life despite the interference of symptoms had to be overcome in the first year for HNC survivors [6]. Thus, not surprisingly, the prevalence of depression among HNC patients following radiation is high (63%) [7]. Patients have expressed experiencing anxiety and stress throughout their cancer treatment [8]. While HNC survivors adapt to the constraints of everyday life, they still have unmet needs, and therefore, there is a need for quantitative measures to assess the wellness levels of these patients.

Quality of life (QoL) scores from validated surveys are increasingly recognized as important tools to address the challenges faced by HNC patients. QoL encompasses various concepts, including physical functioning, role—physical, bodily pain, general health perceptions, vitality, social functioning, role—emotional, and mental health [9]. Moreover, significant relationships exist between specific QoL measures before treatment and survival or functional outcomes among HNC patients [10]. Thus, the specific items in the QoL measurement assist clinicians in addressing the daily life challenges faced by this patient cohort. Recently, survivors of HNC have identified pain, physical fitness, and fatigue as important indices influencing QoL during recovery [11]. Furthermore, other studies have found that strong clinically meaningful associations exist between social functioning, such as social eating, depression, and longitudinal QoL [12]. By focusing on multiple issues experienced by HNC patients in each cohort, the management of the symptoms by hospitals must cover various clinical areas to improve the patient’s journey as a survivor.

The quantity of academic articles related to QoL among HNC patients is rapidly increasing, making it challenging to keep up with all the published articles. Bibliometric analysis is a study of the statistical properties of scientific literature, which involves analyzing various characteristics of published articles, including author productivity, collaboration, impact, and research trends [13]. This method employs tools such as graphs, maps, and network diagrams to visualize the results, facilitating easy interpretation. By providing a structured analysis of a large body of information, bibliometrics help identify trends over time, research themes, shifts in disciplinary boundaries, prolific scholars and institutions, and the overall landscape of existing research [14]. This comprehensive understanding can inform policymakers, scientists, and stakeholders about emerging problems and potential collaborations among scientific groups [15]. Using ranking systems, bibliometric analysis enables decision-makers to quantify and evaluate existing research. Therefore, analyzing the QoL of HNC patients is essential for gaining a comprehensive overview, identifying knowledge gaps, and planning future contributions to the field [16].

To the best of our knowledge, no bibliometric analysis focusing on QoL among HNC patients has been conducted. Such an analysis would highlight developing trends, common research patterns, collaboration networks and forecast future directions. Therefore, our objectives are to (1) identify significant journals, institutions, authors, and countries within this research area and establish networks among them and (2) synthesize crucial keywords used in these articles.

## 2. Materials and Methods

In February 2023, the raw data was collected from the Web of Science Core Collection (WOS-CC). To ensure inclusiveness, the search strategy encompassed subtypes of head and neck cancer (HNC). The search strategy used in this study was as follows: (TI = (quality of life)) AND TI = (“head and neck cancer” OR “oral cancer” OR “paranasal sinus cancer” OR “nasal cavity cancer” OR “sinonasal cancer” OR “nasopharynx cancer” OR “oropharyngeal cancer” OR “hypopharyngeal cancer” OR “laryngeal cancer”). The inclusion criteria for articles were as follows: (1) published between 2013 and 2022, (2) written in English, and (3) not review papers or letters. Figure 1 depicts the PRISMA flowchart for data extraction.

### 2.1. Performance Analysis and Science Mapping

#### Overview of the Articles and Identification of Top Journal

For performance analysis, this study employed Rstudio v.4.0.2 software (R Studio, PBC, Boston, MA, USA) with the bibliometric R-package accessed on 7 February 2023 (http://www.bibliometrix.org), and for science mapping, VOS viewer version 1.6.18 (Centre for Science and Technology Studies, Leiden University, Leiden, The Netherlands) was employed [14,17]. The data were exported to Biblioshiny and analyzed using its web features. The trend of local extracted publications and the average total citations per article were measured for each year, and the global trend of publications graph was created using Microsoft Excel 2020. The number of publications was used to identify the most productive journals, and Bradford’s Law was applied to identify core journals. Bradford’s Law suggests that a small number of core journals contribute to the majority of citations in a field [18].

### 2.2. Identification of Top Institutions, Authors and Countries with Collaboration

The top 10 most productive institutions and authors were ranked based on percentage of papers produced. The relationships between institutions and authors were visualized, where closer proximity indicates a higher likelihood of citing the same publications. The percentage of articles from each country was used to rank the most productive country, while the percentage of multiple-country production (MCP) was measured for the top 10 countries. The country collaboration network was mapped using software that linked countries based on the number of publications produced.

### 2.3. Keywords Analysis

The co-occurrence networks of keywords were analyzed using VOS viewer, and the average normalization citation (ANC) value was overlaid. The ANC represents the average normalized number of citations received by documents published by a source, author, organization, or country. The keywords were clustered using rainbow colors, where cold colors represent research activities with fewer average normalized citations than hot colors. The timeline of keyword occurrence and peak citation years was depicted using a line graph and bubbles, respectively. The development and relevance degrees of emerging themes were analyzed based on centrality and density. Centrality reflects the theme’s relevance in the overall study area, while density indicates the theme’s development. The thematic map was plotted using authors’ keywords, and the Louvain algorithm was used for clustering, as it has shown high effectiveness compared to other algorithms.

## 3. Results

### 3.1. Overview of the Articles and Impactful Journals

This study aimed to review the QoL studies among HNC patients published from 2013 to 2022. A sample of 444 relevant studies published in 177 outlets over the last 10 years was reviewed, written by 2426 authors, with an average of 12.21 citations per document. The majority of authors participated in multi-authored studies (99.9%). The trend of publications each year is depicted in Figure 2a. The total number of publications increased from 2019 to 2022. Eleven core journals were identified based on Bradford’s law, and they were considered excellent options for researchers in this field (Figure 2b). Among the journals, *Supportive Care in Cancer* had the highest production of articles (27, 6.08%). *Head and Neck-Journal for The Sciences and Specialties of The Head and Neck* had the highest number of local citations, with 1065 citations, twice as many as *Supportive Care in Cancer* (Table 1). *Supportive Care in Cancer* is a most productive journal since 2021 and classified in psycho-oncology cluster (Appendix A). Among all 444 studies, paper that have the most global citations and local citations are from Dziegielewski P. T (2013) (DOI: 10.1001/jamaoto.2013.2747, Global citation: 157 times) and Verdonck-De Leeuw I. M, (2014) (DOI: 10.1016/j.radonc.2014.01.002, Local citation: 20 times) (Appendix A). However, the most repeated cited-reference work was from Aaronson et al. (1993) (Appendix A).

### 3.2. Collaboration between Institutions, between Authors, and between Countries

The University of Pittsburgh in the United States was the most productive institution, with the highest number of articles (37, 8.33%) shown in Table 2 and Appendix A showed the cumulative number of each institutions. The top institutions in the Netherlands, Vrije Universiteit Amsterdam and University of Groningen, ranked second (35, 7.88%) and third (28, 6.30%), respectively. For authors, Rogers S. N. had the highest number of articles (19, 4.28%) among other authors, followed by Lagendijk J. A. and Lowe D., who each produced 14 articles (Table 2). The collaboration network between institutions showed a strong collaboration between the top-ranked institution and other Asian universities in China and Taiwan (Appendix A). The collaboration network between authors identified 11 clusters, with the largest cluster (red) exhibiting the strongest collaboration. Notably, Leemans C. R., ranked as the top-4 author, demonstrated the highest number of local citations (57 times) and had an early start in contributing to research in this field compared to most of the top-10 authors (Appendix A).

For countries, the United States accounted for 16.67% of the 74 articles, with most of them being single-country productions (Figure 3a). China and Germany ranked second and third in terms of the number of articles produced (11.26% and 8.10%, respectively). Among the top 10 countries, the United Kingdom had the highest percentage of multiple-country production (38.5%), followed by Canada and Spain (33.3% and 30.0%, respectively) (Figure 3a). The collaboration network between countries revealed that the Netherlands and Germany related strongly, thus forming the largest cluster (red) with other European countries, including Japan and Taiwan (Appendix A). The strongest collaboration was mostly from the US and European countries (Figure 3b). The distribution of publications in the field of HNC was predominantly concentrated in developed countries, while underdeveloped countries with higher prevalence of HNC had fewer publications (Appendix A).

### 3.3. Co-Occurance, Hotspots, and Emerging Keywords

Figure 4a illustrates the correlation between keywords and studies on QoL conducted among HNC patients a decade ago. The complete list of keywords can be accessed in the Appendix A. The co-occurrence network of keywords highlights the tendency of authors to use certain keywords together, as indicated by their proximity. Notable hot topics, such as cancer survivorship, psycho-oncology, and tobacco use, are closely linked at the top of the network. The distance between nodes represents the frequency of co-occurrence between keywords. Furthermore, keywords such as survival and depression demonstrate a higher impact compared to surrounding keywords, with their proximity indicating a stronger association. Malnutrition and intervention also have high frequency and ANC value, indicating their importance for future studies. The complete ANC can be accessed from File S1.

The timeline analysis of important keywords reveals that survivorship, dysphagia, and oral mucositis had peak citations in 2021 (Figure 4b). Survival and dysphagia remained relevant in 2022 as well. In comparison to the years 2012–2016, there has been a shift in focus towards specific aspects of QoL indices, such as dysphagia, oral mucositis, and xerostomia, rather than the general effects of QoL after radiation treatments. Lastly, the thematic map identifies five clusters based on the development and relevance of all keywords. The purple cluster centered around the keyword survivorship is an emerging peripheral topic (Figure 4c). Despite appearing only recently in the literature (starting from 2020), this cluster has been mentioned 50 times. The survival cluster represents a strongly developed but still marginal theme within the research domain. This area is crucial, as it has high ANC value, but its development is faster due to its emergence at an early stage (Density Rank 5). Quality of life related to radiotherapy is a fundamental topic with high frequency (n = 704, Centrality Rank 5), while the cluster related to oropharyngeal cancer after radiotherapy treatment is a well-developed and relevant topic (n = 123, Centrality Rank 4). The centrality and density values and rankings for each domain can be found in Table 3. Among the top 10 authors, Johnson J. T. and Nilsen M. L. emerged as pioneers in the survivorship research theme, as indicated by Appendix A. The frequency of this keyword notably increased after 2018, as depicted in Appendix A. In summary, this study reviewed the quality-of-life studies for head and neck cancer patients over the past decade. It identified top journals, impactful articles, collaborations between institutions, authors, and countries, and important and emerging keywords. The findings provide insights into the research landscape and highlight potential areas for future studies.

## 4. Discussion

This study presents the first bibliometric analysis of publications on the quality of life (QoL) among head and neck cancer (HNC) patients. The study suggests that the most critical area for future studies is the survivorship of HNC patients, while hotspots within the domain include specific subtypes of cancer and symptoms such as dysphagia and oral mucositis. These findings can guide researchers to focus on relevant and meaningful issues rather than exploring redundant or insignificant problems.

The emergence of survivorship as a key research area in the domain of QoL among HNC patients signifies a shift in focus towards understanding and addressing the long-term health and well-being of individuals using QoL measures. Survivorship in cancer refers to a person’s health and well-being beyond the initial treatment phase, encompassing physical, emotional, and social aspects [20]. The importance of QoL has already been recognized in the US, where consensus statements from the American Head and Neck Society and best practices for HNC survivorship suggested that cancer surveillance visits should also include screening and evaluation for toxicities, QoL measures, and health maintenance [21,22]. Tools like HN-STAR, an online health concern elicitation tool, have been developed to support survivorship care planning with primary endpoints of HNC-specific QoL and other outcomes such as patient-centered measures [23]. Understanding symptoms and long-term treatment effects, such as unintentional weight loss, fatigue, and muscle pain, is crucial for developing effective survivorship plans [24]. By exploring and understanding the unique challenges faced by cancer survivors, research can contribute to the development of comprehensive care plans that facilitate communication and coordination among specialty and primary care providers, ensuring the long-term well-being of HNC survivors across different countries.

The keywords used by authors have shifted from overall QoL problems after treatment to more specific problems occurring among this cohort. Specific symptoms such as dysphagia and oral mucositis are commonly observed, and both were highly researched in 2021. Dysphagia, or difficulty swallowing, affects nearly half of HNC patients and can persist for more than five years after diagnosis [25,26,27]. It has a detrimental effect on nutrition and functional outcomes, including speech and eating [28]. Additionally, oral mucositis, characterized by damage to the oral mucosa, contributes to physical impairments in oral function and negatively influences QoL. Studies have shown that patients who develop mucositis following radiotherapy experience lower oral health-related QoL compared to those who do not [29,30]. Oral gels are commonly used to reduce pain, but addressing the barrier function alone may not fully alleviate the severity and frequency of mucositis [31]. Both indices showed the same trend, which is that both of them negatively influence QoL, thus the researcher might want to find a better solution for these problems.

The geographical distribution of research on QoL among patients with HNC indicates a heterogeneous environment with varying contributions from different countries. While the United States emerges as the lead in terms of the number of articles and collaborations, other countries’ substantial contributions have been recognized. For example, the United Kingdom has a higher percentage of articles with authors from different regions compared to other countries, which is a possible strong collaboration for future study. Analysis of different countries’ engagement based on publication output, collaborations, and citation effect reveals a global network of research. Regional patterns of cooperation emerge, such as the United States’ expanding partnership with East Asia, particularly China. Each country’s research output is shaped by factors such as research funding availability, institutional support, academic infrastructure, and regional knowledge. The United States dominated the number of articles and collaborations, which can be attributed to its economic status. This finding aligns with previous studies that found the US to have the highest number of articles in the top 100 most cited articles on HNC [32]. The US collaboration rate has also increased over time, particularly with East Asia and the Pacific, including strong collaborative efforts with China [33]. This collaboration is supported by external funding provided by organizations like the National Noncommunicable Disease and Injury (NCDI) Poverty Commissions in the US, which support cancer research in developing countries [34]. Given that a significant proportion of HNC cases occur in low- and middle-income countries in Asia and the Indian subcontinent, collaboration between the US and other developing countries is becoming stronger [1].

There are a few drawbacks to consider in this study. Relying solely on English-language literature may have resulted in overlooking important studies published in other languages. Additionally, the age of a document can significantly impact citation numbers, as older works may have received more citations due to their longer presence in the public domain.

## 5. Conclusions

In conclusion, the focus of quality of life (QoL) studies among head and neck cancer (HNC) patients has shifted towards specific patient-reported outcomes such as dysphagia, oral mucositis, and xerostomia. Early detection and management of these symptoms are crucial to enhance patients’ QoL during and after treatment. Survivorship has gained significant importance, highlighting the need for studies addressing the perspectives of patients, clinicians, and policymakers. Healthcare practitioners should prioritize survivorship care and develop comprehensive plans to address the long-term needs of HNC survivors. Policy development should allocate resources and establish supportive care services and survivorship programs tailored to HNC patients’ unique requirements. Collaboration between high-resource countries and developing nations with a high HNC prevalence is essential for bridging research gaps and increasing the impact of studies. Future research should focus on interventions and strategies to improve survival outcomes, effective management of specific symptoms and toxicities, and the impact of novel treatment modalities on QoL. Interdisciplinary collaborations can further advance knowledge in this field and contribute to the overall well-being of HNC patients.

## Figures and Tables

**Figure 1 cancers-15-04551-f001:**
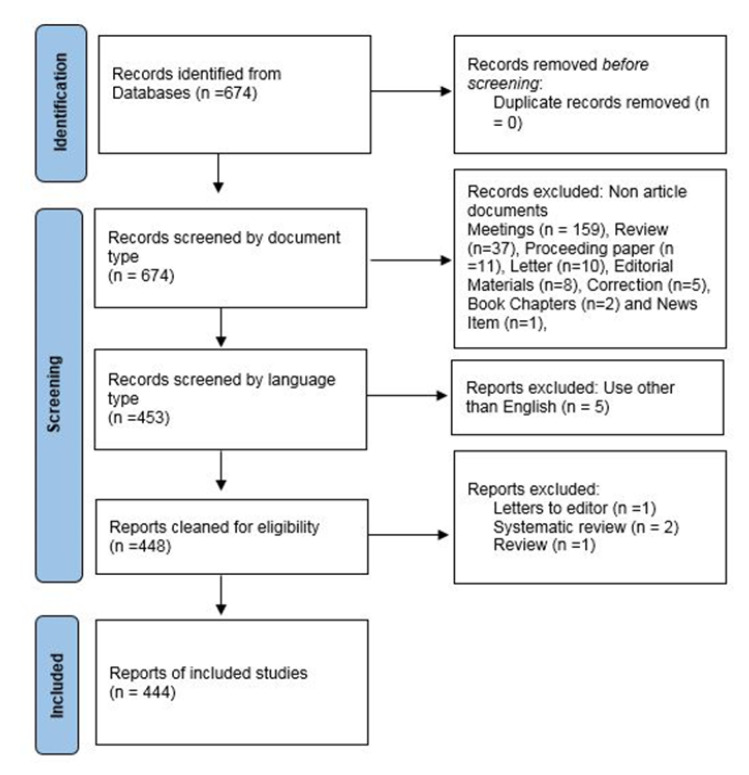
The flow chart of the screening process using PRISMA.

**Figure 2 cancers-15-04551-f002:**
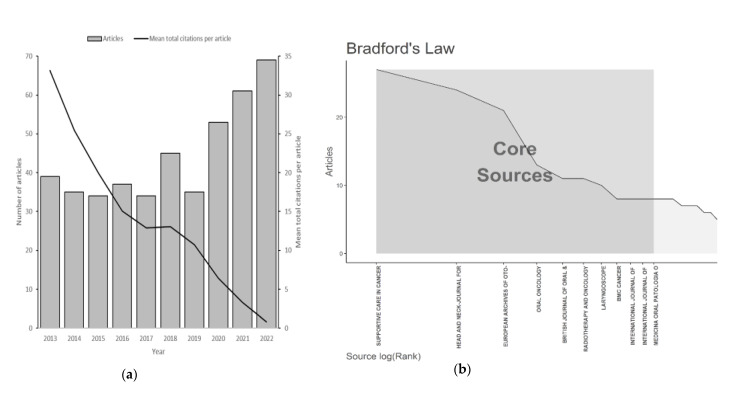
(**a**) Number of articles and average of total citations per articles over the years; (**b**) The plot of Broadford’s Law identified eleven core journals regarding quality of life among head and neck cancer patients.

**Figure 3 cancers-15-04551-f003:**
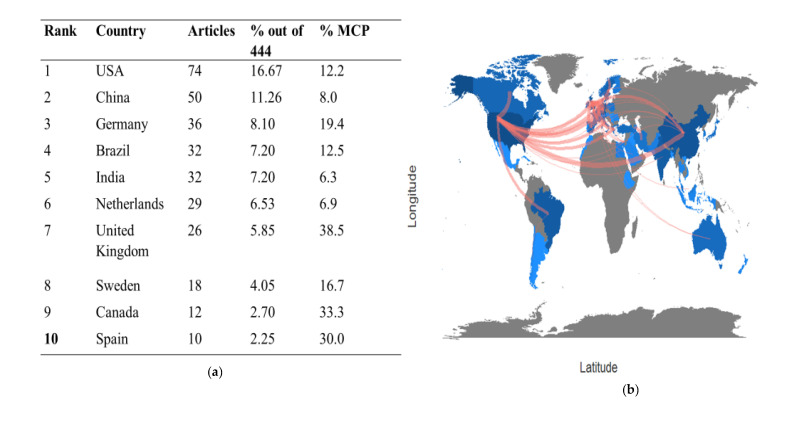
(**a**) The top 10 countries that ranked based on the percentage of articles; (**b**) the geographical maps with collaboration network (red line). The darker the blue shade, the greater the amount of the articles produced.

**Figure 4 cancers-15-04551-f004:**
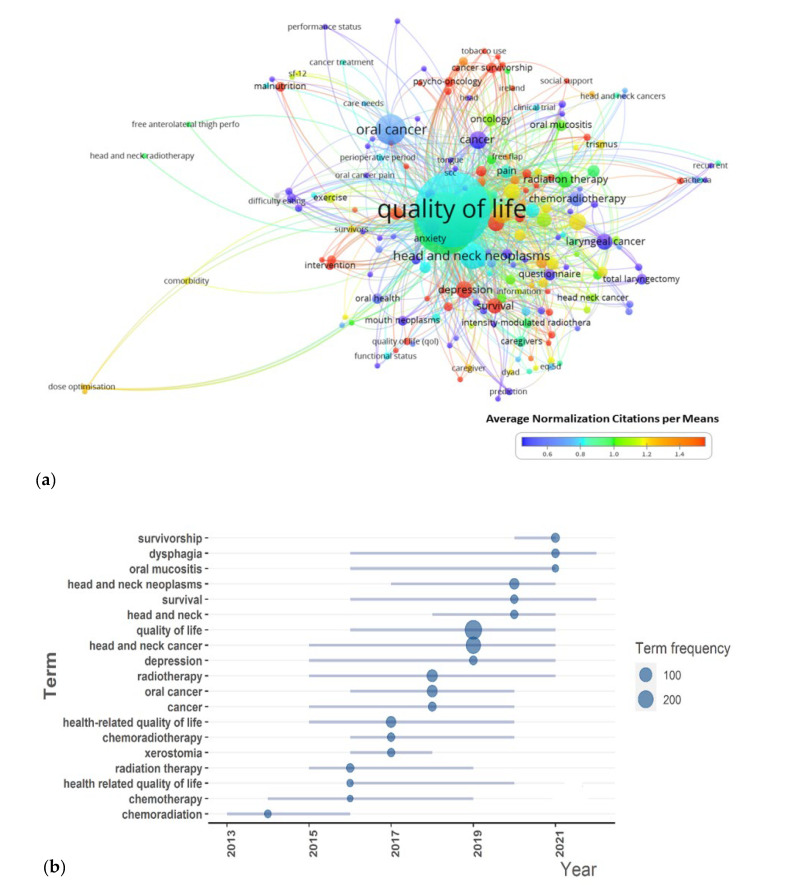
(**a**) The co-occurrence of author keywords from articles. The closer the keywords, the higher the number of occurrences between them. If the keyword had higher ANC than 1.00, it has more impact on the field. (**b**) The timeline of the keywords. Each bubble indicates the peak of frequency used for each keyword, while the line indicates the years it was used. (**c**) The distribution of five major clusters of studies from the dataset.

**Table 1 cancers-15-04551-t001:** The ranking of the journals based on the number of publications and total local citations.

Rank	Sources	Articles	%	TLC
1	Supportive Care in Cancer	27	6.08	453
2	Head and Neck-Journal for The Sciences and Specialties of The Head and Neck	24	5.40	1065
3	European Archives of Oto-Rhino-Laryngology	21	4.72	260
4	Oral Oncology	13	2.92	679
5	Radiotherapy and Oncology	11	2.47	295
6	British Journal of Oral & Maxillofacial Surgery	11	2.47	185
7	Laryngoscope	10	2.25	457
8	International Journal of Radiation Oncology Biology Physics	8	1.80	541
9	Quality of Life Research	8	1.80	183
10	International Journal of Oral and Maxillofacial Surgery	8	1.80	157
11	BMC Cancer	8	1.80	93
12	Medicina Oral Patologia Oral Y Cirugia Bucal	8	1.80	60
13	Oral Surgery Oral Medicine Oral Pathology Oral Radiology	8	1.80	43
14	Otolaryngology-Head and Neck Surgery	7	1.58	179
15	Journal of Cranio-Maxillofacial Surgery	7	1.58	96
16	JAMA Otolaryngology-Head & Neck Surgery	7	1.58	95
17	European Journal of Oncology Nursing	6	1.35	57
18	Indian Journal of Otolaryngology and Head & Neck Surgery	6	1.35	47
19	Acta Oncologica	5	1.13	141
20	Cancers	5	1.13	76

**Table 2 cancers-15-04551-t002:** The top 10 institutions and the top 10 authors based on the percentage of articles.

Rank	Institutions	Country	Articles	%
1	University of Pittsburgh	USA	37	8.33
2	Vrije Universiteit Amsterdam	Netherland	35	7.88
3	University of Groningen	Netherland	28	6.30
4	University of Gothenburg	Sweden	22	4.95
5	University of Michigan	USA	22	4.95
6	University Medical Center Hamburg-Eppendorf	Germany	21	4.73
7	University of North Carolina	USA	21	4.73
8	University of Texas MD Anderson Cancer Center	USA	21	4.73
9	Aintree University Hospitals NHS Foundation Trust	United Kingdom	20	4.50
10	Edge Hill University	United Kingdom	18	4.05
**Rank**	**Authors**	**Country**	**Articles**	**%**
1	Rogers S. N.	United Kingdom	19	4.28
2	Langendijk J. A.	Netherland	14	3.15
3	Lowe D.	USA	14	3.15
4	Leemans C. R.	Netherland	12	2.70
5	Verdonck-De Leeuw I. M.	Netherland	10	2.25
6	Gellrich N.C.	Germany	8	1.80
7	Johnson J. T.	USA	8	1.80
8	Kanatas A.	United Kingdom	8	1.80
9	Jansen F.	Netherland	7	1.58
10	Nilsen M. L.	USA	7	1.58

**Table 3 cancers-15-04551-t003:** The value of density and centrality of each cluster represented in Figure 4c. Centrality refers to the importance of the theme in the entire research area, and density is a measure of the theme’s development [19].

Cluster	Centrality	Density	Rank Centrality	Rank Density	Cluster Frequency
Red	0.603631	13.46375	5	2	704
Blue	0.321137	17.29874	4	3	123
Green	0.257075	19.41352	3	4	70
Purple	0.186034	11.8413	2	1	50
Orange	0.137372	20.85649	1	5	49

## Data Availability

Data are available in Appendix A. Further data are available from the corresponding author on reasonable request.

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
