# Peer review of "Quality of Life for Head and Neck Cancer Patients: A 10-Year Bibliographic Analysis"

_cancers, 2023, doi:10.3390/cancers15184551_

Round 1

Reviewer 1 Report

The objective of this study is to perform a bibliometric analysis focused on the quality of life of head and neck cancer patients, developing trends, research trends, collaborative networks, and anticipate future directions. 

An easy to read and interesting study for researchers in the field. 

Here are my contributions:

  • Why does it say on line 32 until 2022 and then on line 97 the year 2023? (Line 103, 142)

  • Why did you only search in web of science? There are many other quality search engines.

  • Line 223, why does the explanation of the abbreviation appear again? (Line 306)

  • Line 232, in the discussion and even more so at the beginning of the discussion it does not make sense to explain the objectives of a bibliometric analysis and its benefits, you should discuss the results obtained in your study, not talk about the methodology used and its properties.

Reviewer 2 Report

The title of the manuscript is good. English language has good quality. Figures and tables that are located in the figures need some adjustments. Some sections of the manuscript need some changes. There are some modifications that need to be exerted in the citations.

1. Please reform the section "Abstract" based on order below:

Write briefly about:

+ the importance of head and neck cancer (HNC)

+ the importance of your work

+ material and method

+ results

+ conclusions

2. The authors have conducted bibliometric analysis. Please explain why you have not inserted the table that contains how many clusters, items and ... did you obtain from

your bibliometric analysis?

3. The authors have performed only Network visualization. Why you have not performed Overlay visualization?

4. About Figures

All of the figures that contain text should be bigger so that their text can be easily readable. Please reform all of your figures based on this comment.

5. All multipple and middle sentence references in all over the manuscript should be reformed.

6. About the tables that are located in the figures.

Please separate these tables and write about each one of them separately, either in the part "Results" or in the part "Discussion"

7. About the part "Discussion"

Please rewrite this part according to notes below:

First: categorize all of your results based on their importance (from the most important one to the least important)

Second: after that, turn each one of your results into some subheadings

Third: after that, discuss about them one by one

Forth: make comparisons between your results and the results of other similar and relevant surveys

8. This is important that the authors compare their findings with the results of prior similar surveys in the part "Discussion".

9. Please check and adjust the "Reference list" based on the regulations of reference list

of journal. (Titles, doi, the name of journal and ... )

Round 2

Reviewer 1 Report

Congratulations to the authors

Reviewer 2 Report

Thanks for considering my comments.